# The Acari Hypothesis, II: Interspecies Operability of Pattern Recognition Receptors

**DOI:** 10.3390/pathogens10091220

**Published:** 2021-09-21

**Authors:** Andrew C. Retzinger, Gregory S. Retzinger

**Affiliations:** 1Department of Emergency Medicine, Camden Clark Medical Center, West Virginia University, Parkersburg, WV 26101, USA; 2Department of Pathology, Feinberg School of Medicine, Northwestern University, Chicago, IL 60611, USA; gretzing@nm.org

**Keywords:** acari hypothesis, ticks, mites, IgE, vector, ixoderin, fibrinogen-related protein (FReP), interspecies operability

## Abstract

Hypersensitivity to galactose-α-1,3-galactose (α-gal) is an informative example of a pathologic IgE-mediated process. By way of their saliva, ticks are able to sensitize humans to tick dietary elements that express α-gal. Mites, which along with ticks constitute the phyletic subclass Acari, feed on proteinaceous foodstuffs that represent most, if not all, human allergens. Given: (1) the gross nature of the pathophysiological reactions of allergy, especially anaphylaxis, (2) the allergenicity of acarian foodstuffs, and (3) the relatedness of ticks and mites, it has been hypothesized that human-acarian interactions are cardinal to the pathogenesis of allergy. In this report, a means by which such interactions contribute to that pathogenesis is proposed.

## 1. Background/Introduction

Although there exists a link between human allergens and acarians [1,2,3,4,5,6,7,8,9], mere contact between humans and acarians cannot account for allergic inflammation. After all, mites are ubiquitous within the modern human habitat, existing as synanthropic organisms within human dwellings [10], as parasites of human foodstuffs [11,12,13,14,15,16], and as permanent ectoparasites on human skin [17]. In addition, although such ubiquity ensures near-continuous human-acarian interaction, the low prevalence of acarian-targeted adaptive responses seems incongruous. Nevertheless, the localization and scale of IgE-mediated mechanical reflexes on epithelial surfaces argue persuasively that the ectoparasite is, in fact, the target of the adaptive response [18]. Inasmuch as the principle acarian threat to mammalian immunity is vectorial, it seems likely this threat drives the mammalian adaptive response. More precisely, the mammalian immune system recognizes and uses pathogen-bound acarian operators, thereby linking the pathogen to the vector and directing an adaptive anti-vector response.

Mammalian innate immunity is orchestrated by pathogen recognition receptors (PRRs) that identify pathogen-associated molecular patterns (PAMPs) expressed on the membranous surfaces of microorganisms. Binding of PRRs to PAMPs prompts a variety of immune phenomena, including complement activation and phagocytosis [19,20,21]. Sans an adaptive arm, acarian immunity functions in similar fashion. The humoral defense of acarians, like that of mammals, is based on the activity of PRRs and effector molecules, including lectins, complement-like molecules and antimicrobial peptides [22]. Relatedly, the cellular immunity of acarians involves leukocyte equivalents, i.e., hemocytes, that phagocytize, encapsulate, and digest foreign elements in opsonin-dependent fashion [22,23].

The coexistence of pathogens and housekeeping opsonins in acarian salivary glands and gut [22,24] makes it a certainty that complexes comprised of both are transmitted in acarian saliva and/or stool when an acarian interacts with a human, Figure 1.

Because molecular elements derived from dietary materials also populate acarian salivary glands and gut, they, too, undoubtedly admix with the other reactants, forming higher order complexes. Such complexes may be attributed ‘inappropriately’ to the acarian itself, Figure 2, prompting, ultimately, an adaptive response to the dietary materials rather than to the acarian. As elaborated next, homologies between certain acarian and human immune effectors provide clues as to how an adaptive, anti-vector response—whether felicitous or specious—might be achieved.

Although the immune system of mites is not yet well-characterizedthat of ticks is, especially that of ticks of the genus *Ixodes,* including *I. ricinus*, a causative agent of α-gal hypersensitivity in Europe [26]. Contained within both saliva and hemolymph of these ticks are ixoderins [27], well-described members of a family of acarian PRRs that function as opsonins [24]. Analysis of the genome of *Ixodes* ticks reveals 27 genes encoding three groups of ixoderins, i.e., A, B and C [24]. The ixoderins are well-conserved, with homologs identified in other tick genera, including *Amblyomma* [28], the tick responsible for α-gal sensitization in the United States. In *Ornithodoros moubata*, the opsonin, Dorin M, is a homolog of Ixoderin A [24,29,30].

The ixoderins have significant homology with ficolins, Figure 3 and Table 1, a family of human PRRs. Both are members of a larger group known collectively as fibrinogen-related proteins (FRePs) [31], the shared feature of which is a C-terminal domain homologous with the C-terminus of the γ-chain of the eponymous fibrinogen. Indeed, the fibrinogen-related domain (FReD) of Ixoderin A (aa_44–271_) is more nearly identical to that of Ficolin 1 (aa_109–326_, 42.54%) than it is to that of either Ixoderin B (aa_47–276_, 30.04%) or Ixoderin C (aa_231–463_, 31.12%). The FReD confers to FRePs the ability to recognize and bind to interfacial PAMPs [31]. Some invertebrate FRePs have lectin activity, others have immunoglobulin-like domains that facilitate recognition of specific proteins [31,32]. Because FRePs opsonize and agglutinate invasive microorganisms, their primary function in invertebrates appears to be defensive [33]. Twenty-four human FRePs have been identified [34], and just as do invertebrate FRePs, many/all appear to play a role in innate immunity [34,35,36,37,38,39]. Both ficolins and ixoderins are oligomeric proteins, with higher order oligomerization integral to biologic activity [40,41].

As one might expect, ficolins and ixoderins are similar functionally in that they both bind to, and facilitate the phagocytosis of, shared pathogens [24,39]. By virtue of their affinity for carbohydrates, both are categorized as lectins [24,35,38,39]. Their mirrored functionality is further supported by similarity of tissue expression [24,39], as seen in Table 1. Per the model espoused herein, oligomerization and lectin activity have special relevance: oligomerization, because it facilitates complexation of multiple molecular species, and lectin activity, because it confers specificity to that complexation. It comes as no surprise that antigen glycosylation influences allergenicity [42,43,44,45,46,47].

## 2. Discussion

Given the striking similarities of acarian and mammalian immune effectors, it is reasonable to propose the human immune system recognizes acarian PRRs, using them to direct responses against both the acarian, as vector, and its pathogenic payload. In fact, interspecies immune signaling applicable to humans and acarians has already been demonstrated. The well-described PRR and allergen Der p 2 from the common dust mite, *Dermatophagoides pteronyssinus*, shares structural and functional homology with human MD-2 [48]. In humans, MD-2 complexes with lipopolysaccharide (LPS) and LPS-binding protein (LPBP) to activate Toll-like receptor 4 (TLR4)/CD-14 expressed on leukocytes [49,50]. Der p 2 can substitute for MD-2, facilitating the binding of TLR4 to LPS [48]. Analogous interoperabilities likely exist for other structurally- and functionallyrelated molecules, e.g., human ficolin and acarian ixoderin, of phyletically distant species.

The presence of an acarian PRR on a pathogen surface conveys at least two meaningful bits of information to the mammalian system. Firstly, it signifies the underlying surface is foreign. Secondly, it informs that the source of the foreign surface is an acarian vector. This latter, in turn, enables an adaptive anti-vector response directed against the acarian and/or any vector-associated material perceived by the system to be acarian. According to this scheme, potential allergens will be ones that participated—either as intended or otherwise—in pathogen complexation. In fact, this has been shown for some acarian proteins. Der p 2 and Der p 7 are analogs of MD-2 and LPBP, respectively [48,51]. Inasmuch as MD-2 and LPBP bind LPS, Der p 2 and Der p 7 are likely bound to gram-negative bacteria in tick saliva and stool. The implication is that mammals with an IgE-mediated response against Der p 2 or Der p 7 had previously been exposed to gram-negative bacteria transmitted in the saliva or stool of *D. pteronyssinus*.

## 3. Closing

The Acari hypothesis derived from a sense that both the scale and the localization of IgE-mediated reflexes is most applicable to gross targeting of acarians, i.e., mites and ticks [18]. Given their polyphagous nature, acarians are positioned to incorporate a very significant number of dietary proteins into allergenic complexes. That said, humans are parasitized by many phyletically-distant ectoparasites, some of which are vectors. Accordingly, any ectoparasite with PRRs similar to those of man might influence human immunity by the mechanism proposed here. Review of phylogenic emergence estimates of human ectoparasites [52,53,54,55,56,57,58] and of IgE [59], however, indicates most ectoparasites could not have supplied the evolutionary pressure leading to the formation of IgE, Figure 4. The Acari, on the other hand, predate emergence of IgE [58]. The only other ectoparasites predating IgE are helminths [52]. Interestingly, persons living in coastal communities where undercooked fish are consumed often have allergic responses against both fish and the piscine helminth, *Anisakis simplex* [60,61].

Increasingly, research is linking IgE-mediated allergy to defense against venoms [62,63]. Because some venoms are dietary foodstuffs of acarians [64], e.g., *Varroa destructor* and *Apis mellifera*, their inclusion in allergenic complexes of the sort proposed here is not unexpected. Furthermore, many acarians produce venoms [65,66], the paralytic activity of which directly opposes mammalian IgE-mediated mechanical reflexes. Because IgE-mediated reflexes exist to defend against acarian ectoparasitism, evolution of such venoms by acarians seems a natural evolutionary response. That the anti-acarian response evolved means of managing these venoms is fitting. Finally, because acarians are close relatives of other venomous creatures, e.g., scorpions and spiders, it is conceivable that defenses against acarian venoms protects against those of closely-related species.

One means by which to test the validity of the Acari hypothesis might be retrospective epidemiological studies targeting formula-consuming pediatric patients newly diagnosed with a milk allergy. The benefits of this particular patient population are a restricted diet and a largely supervised and controlled environment. Careful analysis of both formula stores and storage sites might yield culpable acarians, effectively relating acarian exposure to allergy. Subsequent testing and confirmation of IgE-mediated immunity to elements of the identified acarian would then imply causality.

As another means by which to test the Acari hypothesis, one could useIgE knock-out rats and the tropical rat mite, *Ornithonyssus bacoti*, a well-described vector of several rat pathogens. Studies comparing acarian-borne pathogen transmission rates between wild-type rats and IgE knock-outs would likely be very informative.

The model described herein provides rationale relevant to the nature of allergic inflammation. If, as proposed, the binding of a foreign FReP, e.g., an ixoderin, to a molecular species directs the generation of an IgE class antibody, then it is tempting to speculate that the binding of a native FReP, e.g., a ficolin or even fibrinogen, to a molecular species might direct the generation of a different class of antibody, the theoretical and practical implications of which would be significant.

## Figures and Tables

**Figure 1 pathogens-10-01220-f001:**
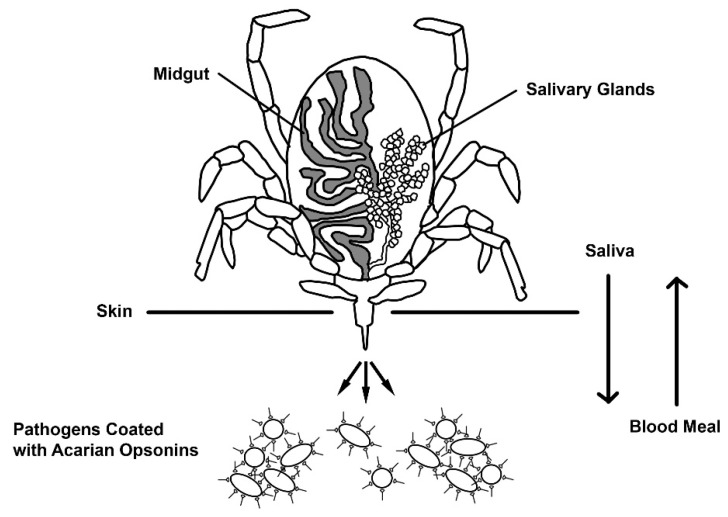
An acarian transmits its opsonized pathogens. The image shows only one means by which such pathogens might be transmitted, here, via hematophagous route. Other means of transmission might include inhaling or consuming the opsonized pathogen or, alternatively, the living vector, allowing for continuous deposition of opsonized pathogens on an epithelial surface. Adapted from [25].

**Figure 2 pathogens-10-01220-f002:**
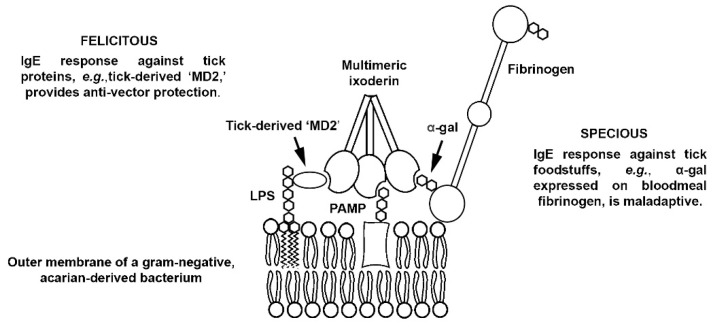
Pictorial representation of a proposed quaternary complex formed between an acarian pathogen (e.g., a gram-negative bacterium), an acarian ‘tag’ protein (e.g., tick ‘MD2′), an acarian opsonin (e.g., multimerized ixoderin) and an acarian foodstuff (e.g., fibrinogen expressing α-gal). The adaptive immune responses elicited by the complex include: (1) felicitous anti-vector, i.e., anti-acarian, IgE directed against the tag protein, (2) specious IgE directed against the foodstuff, (3) felicitous Ig directed against the pathogen. See text for details.

**Figure 3 pathogens-10-01220-f003:**
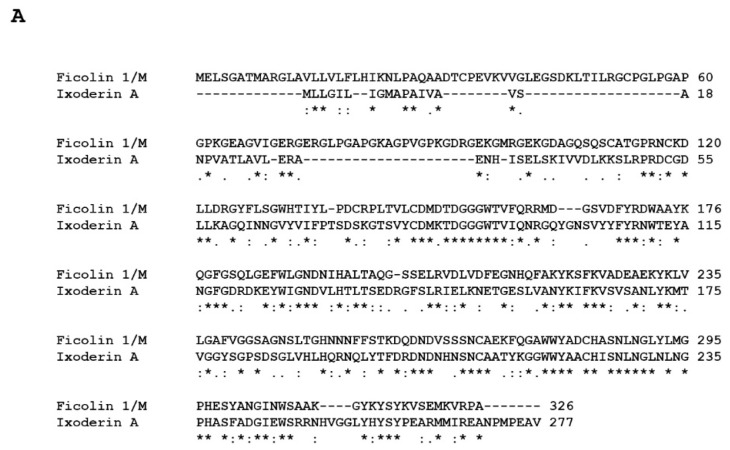
Sequence alignments of human ficolins and acarian ixoderins. Ficolins are paired with ixoderins with which they have greatest homology. (See Table 1). (**A**), Ficolin 1/M (O00602) and Ixoderin A (I6LAP5); (**B**), Ficolin 2/L (Q15485) and Ixoderin C (GCJO01000224); and (**C**), Ficolin 3/H (O75636) and Ixoderin B (Q5IUW6). An * (asterisk) indicates a position at which a single residue is fully conserved, a : (colon) indicates a position at which a substitution has strongly similar properties, and a . (period) indicates a position at which a substitution has weakly similar properties.

**Figure 4 pathogens-10-01220-f004:**
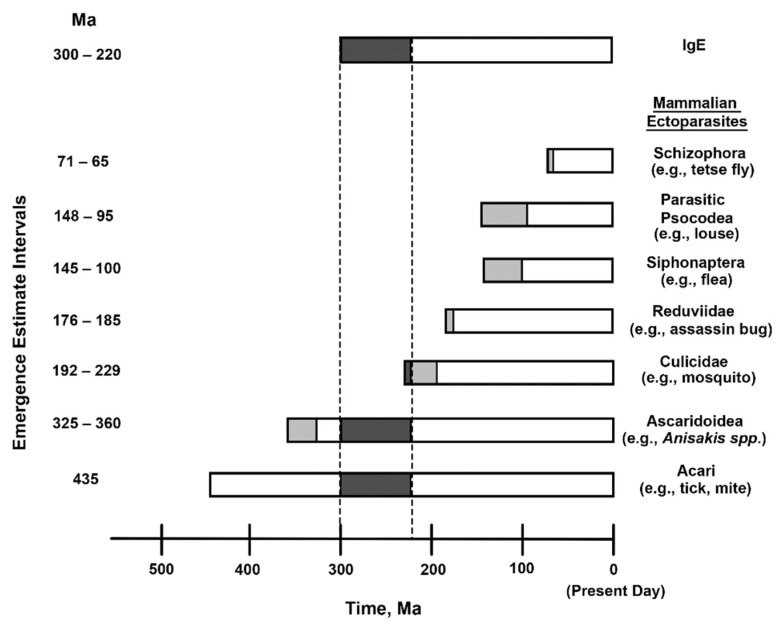
Relative emergences of IgE and ectoparasites of mammals. Intervals of estimated emergences are given both in the left-hand column and in the light gray shaded areas of the horizontal bars. Dashed lines and dark gray shading indicate temporal overlaps of the emergence estimates. Representative ectoparasites are given in the right-hand column.

**Table 1 pathogens-10-01220-t001:** Homologies Between Ficolins of Homo sapiens and Ixoderins of Ixodes ricinus.

Protein ^1^	Sequence Length ^2^	Identical Positions	Similar Positions	Global Identity, ^2^ %	Tissue Expression
Ficolin 1 (O00602)	326	110	80	32	Monocytes, lungs and spleen
Ixoderin A (I6LAP5)	277				Hemocytes and Malpighian tubule
Ficolin 2 (Q15485)	313	100	83	21	Liver (serum protein)
Ixoderin C (GCJO01000224)	463				Ubiquitous expression in all tissues; highest in gut and trachea
Ficolin 3 (O75636)	299	71	89	21	Gallbladder and lungs
Ixoderin B (Q5IUW6)	286				Salivary glands

1 Uniprot primary accession number or European Molecular Biology Laboratory accession number. 2 Including signal peptide.

## Data Availability

Not applicable.

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
