# Peer review of "The Acari Hypothesis, II: Interspecies Operability of Pattern Recognition Receptors"

_pathogens, 2021, doi:10.3390/pathogens10091220_

Round 1
Reviewer 1 Report
I have no further comments on this manuscript.
Reviewer 2 Report
The authors have addressed all issues previously raised and so I am pleased to recommend that this interesting hypothesis article be published.
This manuscript is a resubmission of an earlier submission. The following is a list of the peer review reports and author responses from that submission.
Round 1
Reviewer 1 Report
The shown hypothesis does not have any scientific evidence or result that can confirm future directions.
Reviewer 2 Report
Retzinger and Retzinger submitted an interesting hypothesis article describing effects of possible cross talk between acarian and mammalian PRRs which may lead to development of immunopathological effects. The hypotheses is written in a consistent style and I have not found any serious faults reading the manuscript. Nevertheless I would like to encourage the authors to discuss several topics in deeper details:
1) Please provide more details about mechanism standing behind evolutionary conservation of the mentioned PRR between araci and mammals. Are these PRRs missing in other taxa?
2) Please suggest several experiments which could prove your hypotheses (apart der p 2/7).
Reviewer 3 Report
The authors present an interesting hypothesis about the origin of IgE-mediated allergy, which is particularly relevant given the increase in cases of tick bite-related a-gal hypersensitivity in recent years. This is an interesting and well-researched article, which will be of interest to readers of this journal. The Discussion/conclusions could be improved by including a few lines on future work that needs to be done to prove the hypothesis, e.g. what additional experimental results are needed to confirm this phenomenon.
Otherwise, besides the few specific comments below, I can find little that would improve this work.
Table 1. As these sequences are from Ixodes ricinus, the legend should say Ixodes ricinus rather than Ixodes spp. Spelling - Malpighian tubules
References - There are two references numbered 1, and this has affected the numbering of subsequent references throughout the manuscript.